# Macrophage Cholesterol Efflux Downregulation Is Not Associated with Abdominal Aortic Aneurysm (AAA) Progression

**DOI:** 10.3390/biom10040662

**Published:** 2020-04-24

**Authors:** Marina Canyelles, Mireia Tondo, Jes S. Lindholt, David Santos, Irati Fernández-Alonso, David de Gonzalo-Calvo, Luis Miguel Blanco-Colio, Joan Carles Escolà-Gil, José Luís Martín-Ventura, Francisco Blanco-Vaca

**Affiliations:** 1Servei de Bioquímica, Hospital de la Santa Creu i Sant Pau, IIB Sant Pau, 08041 Barcelona, Spain; mcanyelles@santpau.cat (M.C.); mtondo@santpau.cat (M.T.); 2Departament de Bioquímica i Biologia Molecular, Universitat Autònoma de Barcelona, 08041 Barcelona, Spain; 3Centre of Individualized Medicine in Arterial Disease (CIMA), Department of Cardiology, Odense University Hospital, 5000 Odense, Denmark; Jes.Sanddal.Lindholt@rsyd.dk; 4Institut de Recerca de l’Hospital de la Santa Creu i Sant Pau- IIB Sant Pau, 08041 Barcelona, Spain; dsantos@santpau.cat (D.S.); irati.fernandezalonso@gmail.com (I.F.-A.); david.degonzalo@gmail.com (D.d.G.-C.); 5CIBER de Diabetes y Enfermedades Metabólicas Asociadas (CIBERDEM), 28029 Madrid, Spain; 6Institute of Biomedical Research of Barcelona (IIBB)–Spanish National Research Council (CSIC), 08036 Barcelona, Spain; 7CIBER de Cardiovascular (CIBERCV), Instituto de Salud Carlos III, 28029 Madrid, Spain; lblanco@fjd.es; 8IIS-Fundación Jiménez Díaz, 28040 Madrid, Spain

**Keywords:** abdominal aortic aneurysm, cardiovascular disease, cholesterol efflux, HDL, apoA-I, aortic diameter, growth rate, need for surgery, reverse cholesterol transport

## Abstract

Recent studies have raised the possibility of a role for lipoproteins, including high-density lipoprotein cholesterol (HDLc), in abdominal aortic aneurysm (AAA). The study was conducted in plasmas from 39 large size AAA patients (aortic diameter > 50 mm), 81 small/medium size AAA patients (aortic diameter between 30 and 50 mm) and 38 control subjects (aortic diameter < 30 mm). We evaluated the potential of HDL-mediated macrophage cholesterol efflux (MCE) to predict AAA growth and/or the need for surgery. MCE was impaired in the large aortic diameter AAA group as compared with that in the small/medium size AAA group and the control group. However, no significant difference in HDL-mediated MCE capacity was observed in 3 different progression subgroups (classified according to growth rate < 1 mm per year, between 1 and 5 mm per year or >5 mm per year) in patients with small/medium size AAA. Moreover, no correlation was found between MCE capacity and the aneurysm growth rate. A multivariate Cox regression analysis revealed a significant association between lower MCE capacity with the need for surgery in all AAA patients. Nevertheless, the significance was lost when only small/medium size AAA patients were included. Our results suggest that MCE, a major HDL functional activity, is not involved in AAA progression.

## 1. Introduction

An abdominal aortic aneurysm (AAA) is defined as a permanent dilatation of the abdominal aorta above the threshold of a diameter of 30 mm, as determined using imaging techniques [1]. The prevalence of AAA ranges from 4–8% in men and 0–2% in women, based on population screening and large-scale randomized controlled trials [2,3]. The estimated prevalence of AAA in the US is over one million, with approximately half of AAA cases being women, nonsmokers and aged less than 65 years [4]. AAA is generally asymptomatic, and progressive aneurismal dilation is finally associated with the severe consequences of aortic rupture. To prevent AAA rupture, surgical repair is indicated when the aortic diameter exceeds 55 mm. Other than the AAA diameter, factors such as age, sex, body size and image characteristics should be considered in AAA evaluations [5]. For smaller aneurysms (30–50 mm), follow-up to monitor the growth rate is mandatory to estimate the median growth (mm per year) and/or rupture risk [6]. In these patients, there is no definitive pharmacological treatment. Recent studies showed that the use of statins and low-dose aspirin were associated with lower AAA growth rates and decreased progression [7,8]. However, major randomized double-blind trials are scarce, and no official recommendation regarding medical treatment exists in current guidelines [6,9].

The AAA pathophysiology remains unclear, but it is believed to be associated with alterations in the connective tissue in the aortic wall, mainly due to the high proteolytic activity produced by both infiltrating and resident cells, leading to a decrease in the amount of elastin [10]. A large number of exogenous immune cells, including neutrophils, lymphocytes and macrophages, infiltrate into the aortic tissue, eliciting a significant immune inflammatory response in the AAA wall. These inflammatory cells may enhance smooth muscle cell (SMC)-mediated secretion of matrix metalloproteinases, thereby impairing the stability and mechanical properties of the aortic wall, resulting in destruction of the medial extracellular matrix [11]. The AAA wall is also characterized by the presence of cholesterol crystals, which also induce inflammasome activation [12]. In this context, cholesterol accumulation enhances macrophage differentiation toward a pro-inflammatory state [13].

Among well-established risk factors for AAAs, dyslipidemia has grown in importance since a recent genetic meta-analysis and a Mendelian randomization study demonstrated the potential causal association of lipoproteins with this condition [14,15]. In a prospective study cohort of AAA patients, we found significantly decreased apolipoprotein A-I (apoA-I) (the main protein of high-density lipoprotein (HDL) particles) concentration, and plasma HDL cholesterol (HDLc) concentration predicted the aneurysmal growth rate [16]. However, strong evidence indicates that circulating HDLc levels may only represent a surrogate marker of atherogenesis. The ability of HDL to induce macrophage cholesterol efflux (MCE) is considered one of the main atheroprotective functions of HDL [17]. We recently reported that AAA patients showed impaired HDL-mediated MCE, which could be mechanistically linked to AAA. This inverse association was confirmed after adjusting for age, statin use, plasma lipids, apoA-I and HDLc levels [18]. However, this study included only AAA patients with a large aortic diameter (>50 mm). The association between HDL-mediated MCE and AAA progression has never been evaluated. In this study, we aimed to evaluate HDL-mediated MCE in a cohort of AAA patients with small/medium aortic diameters as a tool to test the potential of MCE to predict AAA growth and/or the need for surgery.

## 2. Materials and Methods

### 2.1. Study Design and Participants

All samples were obtained from a Danish cohort derived from the Viborg Vascular (VIVA) trial (URL: http://www.clinicaltrials.gov. Unique identifier: NCT00662480). The trial was approved by the regional ethics committee on Health Research Ethics (M20080025) on 28 March 2008. All the subjects gave informed consent. The study was performed in accordance with the ethical principles set forth in the Declaration of Helsinki. One hundred and fifty-eight male patients aged 64–74 years with different AAA sizes were randomly selected and classified into three groups according to their aortic diameter, which was measured by abdominal ultrasound: a large size group (aortic diameter > 50 mm; *n* = 39, based in the US Aneurysm Detection and Management study), small/medium size group (aortic diameter between 30 and 50 mm; *n* = 81) and control group (aortic diameter < 30 mm; *n* = 38). This subset was selected from a large collection of plasmas from the VIVA trial [19], and HDLc/apoA-I levels as well as other clinical parameters were similar to the complete collection.

The large size group was referred for a computed tomography scan and vascular assessment. The small/medium size group underwent medical monitoring for clinical control to check for diameter expansion. Monitoring consisted of ultrasonographic follow-up of the aortic diameter (a minimum of two follow-ups in a 5-year period) to obtain a linear growth rate per year. Based on the rate, the patients were divided into three subgroups: low progression (growth rate of < 1 mm per year; *n* = 26), medium progression (growth rate between 1 and 5 mm per year; *n* = 29) and high progression (growth rate of > 5 mm per year; *n* = 26). The patients were assigned to surgery according to increases in the aortic diameter and evaluation of clinical parameters.

### 2.2. Lipid, Apolipoprotein and Lipoprotein Analyses

Whole blood samples were collected in Vacutainer^®^ tubes and fractionated by centrifugation at 1300× *g* for 15 min at room temperature to obtain plasma. Plasma was aliquoted into 1.5 mL tubes and kept frozen at −80 °C until analysis. Plasma total cholesterol and triglyceride (TG) concentrations were determined enzymatically using commercial kits and a COBAS 501c autoanalyzer (Roche Diagnostics, Rotkreuz, Switzerland). ApoA-I levels were determined by an immunoturbidimetric assay (Roche Diagnostics). HDLc levels were measured in plasma obtained after precipitation of apoB-containing lipoprotein particles with phosphotungstic acid and magnesium ions (Roche Diagnostics). Low-density lipoprotein (LDL) cholesterol levels were calculated with the Friedewald equation.

### 2.3. Macrophage Cholesterol Efflux Assays

The MCE capacity of apoB-depleted plasma samples (equivalent to 5% of plasma containing mature HDL, nascent preβ-HDL particles and HDL regulatory proteins) was determined using J774.A1 [^3^H]-cholesterol-labeled murine macrophages according to a previously described protocol [18,20]. Briefly, macrophages were seeded and grown for two days in the Roswell Park Memorial Institute (RPMI) growth medium. Macrophages were then incubated for 48 h with a loading medium containing 1 µCi of radiolabeled cholesterol/well. The cells were washed and incubated with a serum-free medium supplemented with fatty acid-free Bovine serum albumin (BSA) for 18 h to allow equilibration of the radiolabeled cholesterol with the intracellular cholesterol pools. After equilibration, the medium was removed, and the cell cultures washed. The macrophages were then incubated for 4 h in the presence of apoB-depleted plasma (equivalent to 5% of plasma), after which cholesterol efflux was determined and expressed as ([^3^H]-cholesterol medium)/([^3^H]-cholesterol cells medium) × 100. The samples were assayed in duplicate in five independent batches using six-well plates. To minimize the effects of intraplate variation, both AAA and control samples were included in each experiment.

### 2.4. Statistical Analysis

Data are presented as mean ± standard deviation (SD) for continuous variables and as frequencies and percentages for categorical variables. A chi-square test was used to compare the categorical data between groups. The normality of the data was analyzed using the Kolmogorov–Smirnov and D’Agostino and Pearson omnibus test. A one-way analysis of variance (ANOVA) test was used to compare the continuous variables, and Tukey’s post-test was used for comparing differences among groups. Correlations between variables were analyzed using Pearson’s correlation analysis. Multivariate lineal regression models were used to explore the association between efflux and the aortic baseline diameter and growth rate, adjusting for potential confounders. A multivariate Cox regression, analyzing tertiles as a categorical variable using the upper tertile as reference, was performed to explore the association between efflux and time to surgery, adjusting for potential confounders. Clinical confounders were chosen based on previous evidence that associated some clinical parameters and statin use with AAA. The statistical software R (http://www.r-project.org) and GraphPad Prism 5.0 software (GraphPad, San Diego, CA, USA) were used to perform all statistical analyses. A *p*-value < 0.05 was considered to represent a significant difference in all the analyses.

## 3. Results

The clinical and plasma biochemical characteristics of the patients and controls are shown in Table 1. The body mass index (BMI) and diastolic blood pressure (DBP) of the AAA patients were significantly higher than those of the control group, whereas the ankle-brachial index (ABI) was lower than that of the control group. No differences among the groups were observed in terms of smoking habits, diabetes, arterial hypertension, a history of cardiovascular events, use of statins and use of low-dose aspirin. As previously reported [16,18], the apoA-I concentrations in the AAA patients at presentation were significantly lower than those in the control group. Only patients with large size AAAs presented with significantly lower total cholesterol concentrations as compared with those in the control group, and this was concomitant with lower concentrations of LDL cholesterol. Patients with small/medium size AAAs had higher concentrations of very-low density lipoprotein VLDL cholesterol and TG. There was no significant difference in HDLc concentrations when this parameter was compared among the three groups.

The clinical and plasma biochemical characteristics of the small/medium size AAA groups, classified as subjects with low, medium and high AAA progression, are shown in Appendix A
Appendix A. No significant differences were found for lipid, apoA-I and lipoprotein levels and almost all of the studied clinical parameters when they were compared among the three AAA progression groups, thereby indicating that these parameters were not related with AAA progression, at least in our sub-cohort of the VIVA trial. The only exception was the lower use of statins and aspirin in the high progression subgroup.

The ability of apoB-depleted plasma to induce MCE was evaluated in all the groups. MCE was impaired in the large size AAA group as compared with that in the small/medium size AAA group and the control group (Figure 1a). However, no significant differences in HDL-mediated MCE capacity were observed when the different AAA progression subgroups were compared (Figure 1b).

A significant linear trend was detected among the indicated groups after a one-way ANOVA analysis was performed (R square = 0.1010; *p* < 0.0001). The association between MCE capacity and AAA remained significant after adjusting for age, BMI and DBP (Appendix A
Appendix A).

Univariate Pearson correlation tests revealed that the MCE capacity correlated negatively with the aortic baseline diameter and BMI and positively with apoA-I and HDLc in all subjects (Appendix A
Appendix A). However, after adjusting for potential confounders, such as age, smoking, BMI, statin use and DBP, the MCE capacity did not correlate with the aortic baseline diameter (Table 2). Furthermore, when Pearson’s correlation tests were performed only in the AAA patients, the significant correlation between the MCE capacity and aortic diameter disappeared, whereas BMI and the apoA-I and HDLc concentrations remained significant (Appendix A
Appendix A).

Moreover, no significant correlation was found between the MCE capacity and AAA growth rate in the small/medium size group (Table 3), even after adjusting for potential confounders (Appendix A
Appendix A). Importantly, in the small/medium size AAA group, the positive associations between MCE capacity and apoA-I and HDLc concentrations remained significant, as well as the negative association with BMI.

A multivariate Cox regression analysis was also conducted across HDL-mediated MCE tertiles to evaluate the association of the MCE capacity with the need for surgery, adjusted for potential confounders (smoking, a history of cardiovascular disease, use of low-dose aspirin, statins or angiotensin-converting enzyme inhibitors, DBP, BMI, lowest ABI and initial AAA diameter). When all the AAA patients were included (small/medium and large size groups), the need for surgical repair hazard ratio was significant after adjusting for potential confounders (Figure 2a and Appendix A
Appendix A). However, this analysis did not reveal significant associations between the MCE capacity and the need for surgical repair when only small/medium size AAA patients were considered (Figure 2b and Appendix A
Appendix A).

## 4. Discussion

Recent reports support the concept that lipoproteins play a role in the pathogenesis of AAAs [14,15,21]. In agreement with our previous results in AAA subjects with large AAA diameter, LDLc and apoA-I were downregulated in the late stages of the disease [18]. However, cholesterol transported by HDL did not change and it is not considered a good surrogate marker of the lipoprotein antiatherogenicity. We recently demonstrated that that HDL-facilitated MCE, one of the potential main surrogate markers of HDL function, was impaired in large size AAA patients [18]. In this study, in a larger cohort of AAA patients, we confirmed that HDL-facilitated MCE was downregulated in large size AAA patients compared with that of controls and small/medium size AAA patients. In a previous study, we also demonstrated oxidative modifications in some apoA-I residues of HDLs isolated from AAA tissue (>50 mm) obtained after surgery [22]. These apoA-I modifications were closely associated with reduced HDL-mediated MCE capacity in vitro and in vivo [22]. Myeloperoxidase-induced modification of apoA-I is mainly responsible for the loss of apoA-I cholesterol acceptor activity in AAAs by affecting the conformational stability of apoA-I and enhancing apoA-I displacement from HDLs and, therefore, catabolism [23]. Thus, oxidative modifications in apoA-I residues could partially explain the downregulation of HDL-mediated MCE capacity, particularly in late AAA stages.

Since we demonstrated that HDL-mediated MCE capacity was impaired in the late stages of AAA, a clinically relevant question is whether HDL-mediated MCE capacity is associated with the progression of the disease in small/medium size AAA patients. In this study, we evaluated the ability of apoB-depleted plasma, which includes mature HDL, nascent preβ-HDL particles and HDL regulatory proteins, to induce MCE in a cohort of AAA patients at different stages of AAA evolution. To our knowledge, our study is the first prospective study to evaluate the association of HDL-mediated MCE capacity with both the AAA growth rate and/or need for surgical repair in small/medium size AAA patients. When we evaluated the MCE capacity in the small/medium size AAA group, there were no differences in AAA progression, at least in terms of aneurysm growth. As the growth rate can be an incomplete reflection of the real progression of aneurysm [24], we also investigated the association of HDL-mediated MCE capacity with the time to surgery. When all the AAA patients were included (small/medium and large size groups), and after adjusting for potential confounders, the need for surgical repair was associated to lower MCE capacity. However, significance was lost when only small/medium size AAA patients were included. This was interpreted as a biased HR estimation caused by the inclusion of large size AAA patients, due to the impaired MCE capacity of these patients, whose surgical repair was already indicated. New cohort analyses with a larger number of events and more uniformity (for example, including only subjects with small AAAs between 30 and 40 mm) would be needed, however, to further confirm this point.

Overall, our results indicate that determining HDL function using MCE as a surrogate is irrelevant in terms of predicting AAA progression and need for surgical repair, as is also the case of HDLc. Indeed, HDLs display a wide variety of pleiotropic effects including antioxidant, anti-inflammatory, anti-protease, anti-thrombotic, anti-infectious, anti-apoptotic and vasodilatory roles that may be involved in AAA development. It should be noted that the injection of reconstituted discoidal HDLs reduced experimental AAA formation [25]. In line with these findings, we previously demonstrated that the injection of an apoA-I mimetic peptide (D4F) and overexpression of the main anti-inflammatory/antioxidant HDL enzyme paraoxonase (PON) 1 inhibited experimental AAA progression [16,26]. Furthermore, the serum activity of PON1 was reduced in a small cohort of AAA patients [26]. These results suggest that other HDL functions beyond MCE, such as their antioxidant and anti-inflammatory properties, may be significant determinants of AAA development and would warrant further study.

The present study has some limitations. We used a single sample collected to measure MCE. It would be interesting to test the variation of MCE in two sequential samples of the same small/medium AAA patients at two time points of their follow-up (at least on those who progress and would not require surgery) to evaluate whether the changes in MCE are correlated to changes in aortic diameter. In addition, it is important to note that to obtain correlation with AAA growth or time to surgery, we previously tested the whole VIVA cohort [27,28], so in this case, lack of power could also account for the obtained negative results. However, considering alpha = 0.05 and power = 80%, it was estimated that a difference of 2% in MCE could be detected by studying a minimum of 20 subjects in each group.

## 5. Conclusions

HDL-mediated MCE capacity was impaired in large size AAA patients but a lower MCE was not associated with the AAA growth rate and/or the need for surgical repair in small/medium size AAA patients, suggesting that this major HDL functional activity may not be mechanistically involved in AAA progression.

## Figures and Tables

**Figure 1 biomolecules-10-00662-f001:**
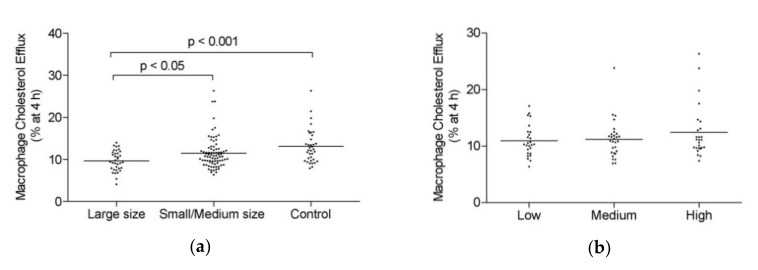
High-density lipoprotein (HDL)-mediated macrophage cholesterol efflux (MCE) capacity. (**a**) Large size abdominal aortic aneurysm (AAA) (*n* = 38), small/medium size AAA patients (*n* = 81) and control group (*n* = 39). (**b**) Small/medium size AAA group based on progression: low (*n* = 26), medium (*n* = 29) and high (*n* = 26) progression. Scatter dot blots are shown, and the line represents the mean. Differences were assessed using Tukey’s multiple comparison test.

**Figure 2 biomolecules-10-00662-f002:**
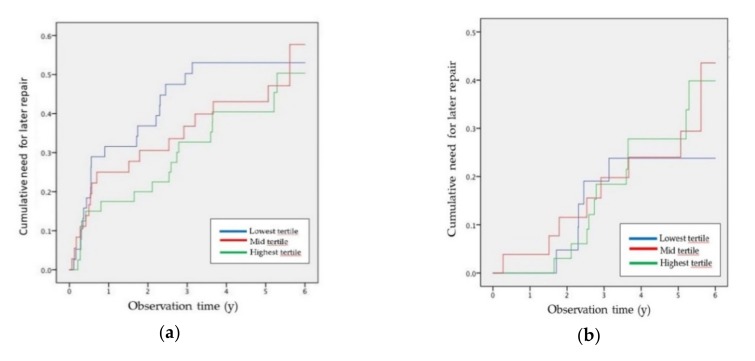
Cox regression of the cumulative need for surgery based on high-density lipoprotein HDL-mediated macrophage cholesterol efflux (MCE) tertiles in all AAA patients (**a**) and in small/medium size AAA patients (**b**). The observation time to surgery is represented in years. The cumulative need for later repair represents the need for surgery over the follow-up. The lowest MCE tertile is shown in blue, the mid tertile is shown in green and the highest tertile is shown in red.

**Table 1 biomolecules-10-00662-t001:** Clinical and plasma biochemical parameters of the abdominal aortic aneurysm (AAA) patients and controls.

Parameters	Large Size (*n* = 38)	Small/Medium Size (*n* = 81)	Control (*n* = 39)	ANOVA or Chi-Square *p*-Value
Age (y)	69.71 ± 2.88	69.65 ± 2.81	68.87 ± 2.69	ns
BMI (%)	28.03 ± 2.42 ^†^	27.41 ± 3.48 ^†^	25.80 ± 2.55	<0.01
Total cholesterol (mmol/L)	4.52 ± 0.71 ^†^	4.87 ± 0.88	5.24 ± 0.77	<0.001
TG (mmol/L)	0.98 ± 0.40 ^#^	1.44 ± 0.66 ^†^	1.10 ± 0.36	<0.001
ApoA-I (g/L)	1.53 ± 0.28 ^†^	1.59 ± 0.32 ^†^	1.80 ± 0.32	<0.001
HDLc (mmol/L)	1.09 ± 0.43	1.09 ± 0.40	1.22 ± 0.42	ns
LDLc (mmol/L)	2.98 ± 0.81 ^†^	3.13 ± 0.90	3.52 ± 0.89	<0.05
VLDLc (mmol/L)	0.45 ± 0.19 ^#^	0.65 ± 0.26 ^†^	0.50 ± 0.16	<0.001
Aortic diameter (mm)	62.52 ± 15.35 ^†,#^	36.35 ± 4.54 ^†^	18.16 ± 2.90	<0.001
DBP (mm Hg)	91.00 ± 13.62 ^†^	88.00 ± 12.30 ^†^	80.97 ± 11.36	<0.01
Lowest ABI	0.99 ± 0.11 ^†^	0.95 ± 0.19 ^†^	1.10 ± 0.09	<0.001
Smoking	8 (2%)	32 (41%)	15 (40%)	ns
Diabetes	3 (8%)	10 (13%)	4 (8%)	ns
Arterial hypertension	15 (40%)	41 (52%)	23 (61%)	ns
Previous CVD	1 (18%)	11 (14%)	7 (3%)	ns
Statin use	16 (42%)	41 (52%)	20 (53%)	ns
Low-dose aspirin	8 (50%)	35 (44%)	19 (22%)	ns

BMI = body mass index; TG = triglycerides; ApoA-I = apolipoprotein A–I; HDLc = High-density lipoprotein cholesterol; LDLc = low-density lipoprotein cholesterol; VLDLc = very low-density lipoprotein cholesterol; DBP = diastolic blood pressure; ABI = ankle brachial index; CVD = cardiovascular disease (acute myocardial infarction, angina or stroke); ANOVA = analysis of variance. Results expressed as mean ± standard deviation (SD). ^†^
*p* < 0.05 compared to the control group, ^#^
*p* < 0.05 compared to small/medium size, ns = non-significant.

**Table 2 biomolecules-10-00662-t002:** Multivariate linear regression of the aortic baseline diameter and macrophage cholesterol efflux (MCE) capacity in all subjects, adjusted for age, body mass index (BMI), smoking, statin use and diastolic blood pressure (DBP).

Coefficients			
Model	Standardized Coefficients	*t*	*p-Value*
	**Beta**		
Age	0.050	0.634	0.527
BMI	0.145	1.622	0.107
Smoke	0.027	0.325	0.745
Statins	0.103	1.273	0.205
DBP	0.275	3.457	0.001
MCE capacity	−0.115	−1.295	0.198
Dependent variable: aortic baseline diameter			

**Table 3 biomolecules-10-00662-t003:** Univariate correlations between high-density lipoprotein (HDL)-mediated macrophage cholesterol efflux (MCE) and abdominal aortic aneurysm (AAA) growth rate, apolipoprotein A-I (apoA-I), high-density lipoprotein cholesterol (HDLc) and body mass index (BMI) in the small/medium size AAA group.

	Growth Rate	ApoA-I	HDLc	BMI
MCE capacity in small/medium size AAA group	0.11 (−0.11–0.32)	0.36 (0.16–0.54)	0.37 (0.16–0.55)	−0.36 (−0.54–(−0.15))
*p*-value	ns	<0.001	<0.001	<0.01

Results expressed as r Pearson coefficient (95% confidence interval), ns = non-significant.

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
