# Peer review of "Macrophage Cholesterol Efflux Downregulation Is Not Associated with Abdominal Aortic Aneurysm (AAA) Progression"

_biomolecules, 2020, doi:10.3390/biom10040662_

Round 1

Reviewer 1 Report

The authors present a timely and adequately written manuscript. The topic is of interest in terms of biomarker identification. the mathods are adequate and conclusive

Abstract:

The abstract leaves out some key messages: was blood used or AAA samples? Please consider revising

Introduction:

- “An aneurysm may be symptomatic or asymptomatic and can be classified according to its diameter.” This sentence is rather unclear: what is classified? Symp/asymp according to diameter? Aneurysm diameter itself? (I am not aware of any consensus classification using diameter)

- “The pathophysiology of an AAA remains unclear, but it is believed to be associated with alterations in the connective tissue in the aortic wall, leading to a decrease in the amount of elastin” apart form this sentence I would recommend to, at least mention inflammation as a major component of AAA pathogenesis.

Material and Methods:

No queries

Results:

figure 2: I would recommend to include a color coded legend in the figure, not only in the subtext

Discussion:

- Please disucss if an evaluation in a prospective cohort with sequentially collectected samples from the same individuum would make sense.

Author Response

Reviewer 1

The authors present a timely and adequately written manuscript. The topic is of interest in terms of biomarker identification. the methods are adequate and conclusive

We thank the reviewer for their important suggestions, which have helped us improve the study. Our responses are presented below, and the changes have been highlighted in the revised version of the manuscript.

 Abstract:

The abstract leaves out some key messages: was blood used or AAA samples? Please consider revising

As requested, we have indicated that the study was conducted in plasmas from AAA patients.

Introduction:

- “An aneurysm may be symptomatic or asymptomatic and can be classified according to its diameter.” This sentence is rather unclear: what is classified? Symp/asymp according to diameter? Aneurysm diameter itself? (I am not aware of any consensus classification using diameter)

We have modified the sentence as follows in the introduction:

AAA is generally asymptomatic and progressive aneurysmal dilation is finally associated with the severe consequences of aortic rupture. To prevent AAA rupture, surgical repair is indicated when the aortic diameter exceeds 55 mm (as stated in the latest European guidelines, new reference 6). However, the large size AAA group was selected based in the US Aneurysm Detection and Management study experience in which subjects with an aortic diameter > 50 mm were referred for a computed tomography scan and vascular assessment (also included in Methods).

- “The pathophysiology of an AAA remains unclear, but it is believed to be associated with alterations in the connective tissue in the aortic wall, leading to a decrease in the amount of elastin” apart form this sentence I would recommend to, at least mention inflammation as a major component of AAA pathogenesis.

 As requested we have also clarified the role of vascular inflammation on AAA pathogenesis and included several relevant references.

The pathophysiology of an AAA remains unclear, but it is believed to be associated with alterations in the connective tissue in the aortic wall, mainly due to the high proteolytic activity produced by both, infiltrating and resident cells, leading to a decrease in the amount of elastin [10]. A large number of exogenous immune cells, including neutrophils, lymphocytes and macrophages, infiltrate into the aortic tissue, eliciting a significant immune inflammatory response in AAA wall. These inflammatory cells may enhance smooth muscle cell (SMC)-mediated secretion of matrix metalloproteinases, thereby impairing the stability and mechanical properties of the aortic walls, resulting in destruction of medial extracellular matrix (Cardiovasc Res. 2011;90:18–27). AAA wall is also characterized by the presence of cholesterol crystals which also induce inflammasome activation (PLoS One. 2010;5:e11765). In this context, cholesterol accumulation enhances macrophage differentiation toward a pro-inflammatory state (Nat Rev Immunol. 2015;15:104–116).

Results:

figure 2: I would recommend to include a color coded legend in the figure, not only in the subtext

As requested, we have included a color coded in the figure 2 legend.

Discussion:

- Please discuss if an evaluation in a prospective cohort with sequentially collected samples from the same individual would make sense.

We have included this sentence as a limitation of our study:

We have a single sample collected to measure MCE, which was later associated to AAA growth and/or need of surgery. It would be interesting to test the variation of MCE in two sequential samples of the same small/medium AAA patients at two time points of their follow-up (at least on those who progress and would not require surgery) to evaluate if changes in MCE are correlated to changes in aortic diameter (lines 320-329).

Reviewer 2 Report

In the present article, the authors aimed to evaluate HDL-mediated MCE in a cohort of patients with small/medium aortic diameters and the potential of MCE to predict AAA growth and/or the need for surgery. The article is written in clear language and the research design is prepared appropriatly. Results are obtained using a large cohort of samples, which ensures the significance of present data. However, the present study has many obstacles. 

Major concerns:

1) The introduction isnt clear - why do we need such research? what is true novelty of the present study?

2) method section is written in a negligent manner. 2.2 and 2.3 sections must be improved. I didnt find any details how collected samples were handled etc.

3) results in table1 (total cholesterol - lowest ABI) should be presented as dot graphs. It is hard to understand clearly. Although authors found some correlations between groups, however some differencies seem to be very small - is it clinically relevant? what is the novelty of the present results?

Most major concern:

The present study is standing basically on negative observations. Therefore, the significance of present content is very low and novelty is uncertain.  

Author Response

Reviewer 2.

In the present article, the authors aimed to evaluate HDL-mediated MCE in a cohort of patients with small/medium aortic diameters and the potential of MCE to predict AAA growth and/or the need for surgery. The article is written in clear language and the research design is prepared appropriatly. Results are obtained using a large cohort of samples, which ensures the significance of present data. However, the present study has many obstacles.

We acknowledge the reviewer for his/her comments and suggestions which have helped us to significantly improve the manuscript. Our responses are presented below, and the changes have been highlighted in the revised version of the manuscript.

Major concerns:

  • The introduction isnt clear - why do we need such research? what is true novelty of the present study?

The pathophysiology of AAA remains unclear. Vascular inflammation is a main hallmark of AAA pathogenesis (Cardiovasc Res. 2011;90:18–27). AAA wall is also characterized by the presence of cholesterol crystals which also induce inflammasome activation (PLoS One. 2010;5:e11765) and cholesterol accumulation also enhances macrophage differentiation toward a pro-inflammatory state (Nat Rev Immunol. 2015;15:104–116). We recently demonstrated that HDL-mediated MCE capacity was impaired in the late stages of AAA (Martínez-Lopez ATVB 2018;38:2750–2754) and this was independent of HDLc levels.These results indicated that HDL-mediated MCE capacity downregulation could be mechanistically linked to AAA. For this reason, as there are not many biological predictors of AAA growth, a clinically relevant question was whether HDL-mediated MCE capacity was associated with the progression of the disease in small/medium size AAA patients. To our knowledge, ours is the first prospective study to evaluate the association of HDL-mediated MCE capacity with both the AAA growth rate and/or need for surgical repair in small/medium size AAA patients. Despite our results confirmed that HDL-mediated MCE capacity was impaired in AAA patients at the late stages, MCE was not associated with the AAA growth rate and/or the need for surgical repair in small/medium size AAA patients, suggesting that this major HDL functional activity may not be mechanistically involved in AAA progression.

We have now included these points in the Introduction and Discussion (lines 66-78 and 283-291).

  • method section is written in a negligent manner. 2.2 and 2.3 sections must be improved. I didnt find any details how collected samples were handled etc.

We agree that the detail of this section can certainly be expanded. More methodological details have now been included in sections 2.2 and 2.3, particularly on sample collection.

  • results in table1 (total cholesterol - lowest ABI) should be presented as dot graphs. It is hard to understand clearly. Although authors found some correlations between groups, however some differencies seem to be very small - is it clinically relevant? what is the novelty of the present results?

There are many variables in Table 1. Translation to Figures will be complicated and cholesterol lipoprotein levels are just confirmatory findings of previous studies, ours and of other authors.

We have tried to explore an aspect of the potential contribution of HDL to AAA development. Any finding, to be clinically relevant would have to find its place in assistance. We have tried to see if MCE would be useful to predict growth and need of surgery in small/medium AAA given that we found impairment of this function in advanced AAA. 

In agreement with our previous results in AAA subjects with large AAA diameter, LDLc and apoA-I were downregulated in the late stages of the disease (Martínez-Lopez ATVB 2018;38:2750–2754). However, cholesterol transported by HDL levels are not good surrogate markers of this lipoprotein antiatherogenicity. For this reason, we directly evaluated the main surrogate marker of HDL function, HDL-mediated MCE capacity. However, no significant differences were found when AAA subjects were subclassified as subjects with low, medium, and high AAA progression (suppl Table 1),.

Due to the reduced number of subjects, lack of statistical power should be considered. However, based in our previous findings (Martínez-Lopez ATVB 2018;38:2750–2754), we estimated that considering alpha=0.05 and power=80%, a difference of 2% in MCE could be detected with a minimum of 20 subjects in each group. However, it is important to note that to obtain correlation with AAA growth or time to surgery, we previously tested the whole VIVA cohort (Vila-Sala et al JAHA 2018, Fernandez-Garcia et al. J Thromb Haemost 2017), so lack of power could account for some negative results found. The problem is that MCE is not a high-throughput technique and, therefore, is difficult to perform analysis of hundreds of samples with low analytical imprecision.

Regarding other clinical parameters, we have reported in the VIVA cohort that DBP was upregulated in AAA, whereas lowest ABI was lower (J. Clin. Med. 2020, 9, 67). DBP is considered a potential risk factor for AAA (Eur J Epidemiol. 2019; 34(6): 547) and patients affected by peripheral artery disease (lowest ABI) seem to be at particularly high risk for AAA development (BMC Surg. 2012; 12(Suppl 1): S17). Although these parameters appear to be affected in AAA subjects, no significant differences were found for these clinical parameters when small/medium size AAA subjects were subclassified as subjects with low, medium, and high AAA progression (suppl table 1).

All these points have been clarified in Results and Discussion (lines 192-198, 263-272 and 325-329).

Most major concern:

The present study is standing basically on negative observations. Therefore, the significance of present content is very low and novelty is uncertain.

As commented above, this is the first prospective study to evaluate the association of HDL-mediated MCE capacity with both the AAA growth rate and/or need for surgical repair in small/medium size AAA patients. We confirmed that HDL-mediated MCE capacity was impaired in AAA patients at the late stages, but MCE was not associated with the AAA growth rate and/or the need for surgical repair in small/medium size AAA patients, suggesting that this major HDL functional activity may not be mechanistically involved in AAA progression.

Even though it would have been preferred to obtain positive results, we disagree that the significance of the present content is very low and the novelty is uncertain. We believe that negative results should be also reported if, as we feel is the case, the study has a solid ground.  This is to our knowledge, the first study trying to apply HDL functionality tests to AAA evolution prediction and we have done it using a well-documented, prospective study, the VIVA trial.

Reviewer 3 Report

The manuscript of Canyelles et al.  describes the quantification of macrophage cholesterol efflux (MCE) from apoB-depleted serum from patients with abdominal aortic aneurysm (AAA).   Different from their previous work which identified a decrease in MCE in AAA patients with large aneurysms, this work is intended to identify the correlation between MCE and small or medium aneurysms.  In addition, this paper hopes to find an association between MCE and the growth/development of the aneurysm, perhaps as an predictive tool.  Overall, the manuscript is thorough, but also a bit underwhelming.  The primary hypothesis will need to be better elaborated and the model for this association stated.  My comments are below.   1) HDL is a multifunctional lipoprotein complex with many described functions.  The authors have tried to imply a cause and effect relationship with MCE and AAA.  Decreased MCE (as measured in this manuscript) is directly correlated with decreased HDL levels (apoA-I, HDL-C).  In addition to decreased MCE, I am sure that the authors could quantify 10 more functional differences of HDL that are decreased in AAA patients' serum.  I understand that the authors are using this as a tool (similar to just measuring HDL-C or apoA-I serum concentrations), but I would like the authors to state this much more clearly.  HDL's anti-thrombotic, anti-oxidative, anti-inflammatory, anti-infectious, anti-apoptotic, and vasodilatory roles may play a more important role in development of AAA.  What does measurement of MCE provide more than just HDL concentration?   2) The authors have been nebulous about the role of HDL or MCE in preventing or protecting from AAA.  The authors should state clearly what their model is and that measuring MCE is a benefit in determining AAA.  Mentioning the study about the administration of D4 peptide reducing or preventing evolution of AAA is helpful (ref. 13, 22, 23).     3) The method for MCE is not stated in the manuscript, but just referenced.  I would ask that, in addition to providing the reference, the authors provide a brief summary of the protocol.   4) A long time ago, a study found that people who smoked, beat their children more often.  One might argue that we should ban smoking because it causes people to beat their children.  However, this was a false comparison.  People who were poor, smoked more, and people who were poor, beat their children more often.  I worry that the authors have also found a false comparison, that MCE is not associated with AAA, but that HDL levels or cardiovascular health is associated with AAA.  What could the authors say and what evidence could the authors provide to rebut my contention that this is a false comparison?

Author Response

Reviewer 3

The manuscript of Canyelles et al.  describes the quantification of macrophage cholesterol efflux (MCE) from apoB-depleted serum from patients with abdominal aortic aneurysm (AAA).   Different from their previous work which identified a decrease in MCE in AAA patients with large aneurysms, this work is intended to identify the correlation between MCE and small or medium aneurysms.  In addition, this paper hopes to find an association between MCE and the growth/development of the aneurysm, perhaps as a predictive tool.  Overall, the manuscript is thorough, but also a bit underwhelming.  The primary hypothesis will need to be better elaborated and the model for this association stated.  My comments are below.

We thank the reviewer for important comments and suggestions, which have helped us improve the manuscript. Our responses are presented below, and the changes have been highlighted in the revised version of the manuscript. We have now clarified in the Introduction that vascular inflammation is a hallmark of AAA pathogenesis (Cardiovasc Res. 2011;90:18–27). AAA wall is also characterized by the presence of cholesterol crystals which also induce inflammasome activation (PLoS One. 2010;5:e11765) and cholesterol accumulation also enhances macrophage differentiation toward a pro-inflammatory state (Nat Rev Immunol. 2015;15:104–116).These inflammatory cells and cellular elements may enhance smooth muscle cells-mediated secretion of matrix metalloproteinases, thereby impairing the stability and mechanical properties of the aortic walls, resulting in loss of smooth muscle cells and destruction of medial extracellular matrix (Cardiovasc Res. 2011;90:18–27). Since HDL-mediated MCE capacity was impaired in the late stages of AAA, independently of HDLc and apoA-I levels (Martínez-Lopez ATVB 2018;38:2750–2754), a clinically relevant question was whether HDL-mediated MCE capacity was associated with disease progression. Despite our results confirmed that HDL-mediated MCE capacity was impaired in AAA patients at the late stages (without altering HDLc), MCE was not associated with the AAA growth rate and/or the need for surgical repair in small/medium size AAA patients, suggesting that this major HDL functional activity may not be involved in AAA progression. We have now included these points in the Introduction and Discussion (lines 66-78 and 283-304).

  • HDL is a multifunctional lipoprotein complex with many described functions. The authors have tried to imply a cause and effect relationship with MCE and AAA.  Decreased MCE (as measured in this manuscript) is directly correlated with decreased HDL levels (apoA-I, HDL-C).  In addition to decreased MCE, I am sure that the authors could quantify 10 more functional differences of HDL that are decreased in AAA patients' serum.  I understand that the authors are using this as a tool (similar to just measuring HDL-C or apoA-I serum concentrations), but I would like the authors to state this much more clearly.  HDL's anti-thrombotic, anti-oxidative, anti-inflammatory, anti-infectious, anti-apoptotic, and vasodilatory roles may play a more important role in development of AAA.  What does measurement of MCE provide more than just HDL concentration? 

As commented above, we evaluated HDL-mediated MCE capacity because we had found that it was impaired in the late stages of AAA, independently of HDLc/apoA-I levels (Martínez-Lopez ATVB 2018;38:2750–2754). We hypothesized, that HDL functionality may be more informative than HDL concentration in plasma. Small fractions of HDL (prebeta HDL for instance) may be responsible of a more important fraction of efflux than that represented by its concentration). As a practical example, decrease in MCE was detected in a previous study (Martínez-Lopez ATVB 2018;38:2750–2754), and confirmed in this one, without concomitant change in HDLc and was independent of apoA-I.

For this reason, we test the potential of MCE to predict AAA growth and/or the need for surgery. This determination can  be directly performed in apoB-depleted plasmas and, in comparison with other antioxidant/antiinflmmatory determinations requiring HDL isolation from plasma through ultracentrifugation, which represents an important limitation. Overall, our results indicate that MCE is irrelevant in terms of AAA progression and need for surgical repair. Therefore, even though this result could not be anticipated. Indeed, HDLs display pleiotropic effects including antioxidant, anti-inflammatory, anti-protease, anti-thrombotic, anti-inflammatory, anti-infectious, anti-apoptotic and vasodilatory roles that may be involved in AAA development.  This would warrant further studies.

We have now included these points in the Discussion (lines 305-310).

  • The authors have been nebulous about the role of HDL or MCE in preventing or protecting from AAA. The authors should state clearly what their model is and that measuring MCE is a benefit in determining AAA.  Mentioning the study about the administration of D4 peptide reducing or preventing evolution of AAA is helpful (ref. 13, 22, 23). 

As commented above, we have highlighted in the Introduction that HDL-mediated MCE capacity was impaired in the late stages of AAA, independently of HDLc and apoA-I levels (Martínez-Lopez ATVB 2018;38:2750–2754) and, for this reason, we aimed to evaluate whether HDL-mediated MCE capacity was associated with the progression of the disease.

D4 peptide prevented experimental AAA evolution, but this could be related with the enhanced ability of D4 to promote HDL-mediated MCE or antioxidative/anti-inflammatory HDL properties. However, it is not unusual to obtain better results in mouse models than in patients.  In this context, we previously published that the main HDL antioxidant enzyme, pon-1, activity was reduced in serum of an AAA Spanish cohort (Burillo et al, 2016 Clinical Science (2016) 130, 1027). The potential of this enzyme activity to predict AAA progression deserve further investigation. This point has been included in the Discussion (lines 315-319).

3) The method for MCE is not stated in the manuscript, but just referenced.  I would ask that, in addition to providing the reference, the authors provide a brief summary of the protocol.  

As requested, methodological details have now been included in section 2.3. Access to more detailed information is referenced.

  • A long time ago, a study found that people who smoked, beat their children more often. One might argue that we should ban smoking because it causes people to beat their children.  However, this was a false comparison.  People who were poor, smoked more, and people who were poor, beat their children more often.  I worry that the authors have also found a false comparison, that MCE is not associated with AAA, but that HDL levels or cardiovascular health is associated with AAA.  What could the authors say and what evidence could the authors provide to rebut my contention that this is a false comparison?

We have tried to see if MCE, as a direct measure of HDL function and not only their concentration, would be useful to predict growth and need of surgery in small/medium AAA given that we found impairment of this function in advanced AAA (a situation in which HDLc did not change).  We estimated that considering alpha=0.05 and power=80%, a difference of 2% in MCE can be detected with a minimum of 20 subjects in each group. Unfortunately, our results indicate that determining HDL function as a mediator of MCE is irrelevant in terms of AAA progression and need for surgical repair. HDLc or apoA-I did not predict evolution either in small/medium AAA . However, it is important to note that to obtain correlation with AAA growth or time to surgery, we previously tested the whole VIVA cohort (Vila-Sala et al JAHA 2018, Fernandez-Garcia et al. J Thromb Haemost 2017), so lack of power could account for some negative results found (now included as limitations of the study, lines 325-329). The problem is that MCE is not a high-throughput technique and, therefore, is difficult to perform analysis of hundreds of samples with low analytical imprecision. Whether other biological HDL functions will provide more information remains to be seen and warrants further study.  

Round 2

Reviewer 2 Report

The manuscript was highly improved according to my comments. The authors gave full and scientifically significant answers to the concerns I had previously. The overview of the present manuscript novelty was fully defended by the authors.